# MicroRNAs and Long Noncoding RNAs as Novel Therapeutic Targets in Estrogen Receptor-Positive Breast and Ovarian Cancers

**DOI:** 10.3390/ijms22084072

**Published:** 2021-04-15

**Authors:** Tushar Singh Barwal, Uttam Sharma, Sonali Bazala, Ipsa Singh, Manju Jain, Hridayesh Prakash, Shashank Shekhar, Elise N. Sandberg, Anupam Bishayee, Aklank Jain

**Affiliations:** 1Department of Zoology, Central University of Punjab, Ghudda 151 401, Punjab, India; tushar101singhbarwal@gmail.com (T.S.B.); uttamsharma1994@gmail.com (U.S.); sonalibazala97@gmail.com (S.B.); ipsapoonam08@gmail.com (I.S.); 2Department of Biochemistry, Central University of Punjab, Ghudda 151 401, Punjab, India; manjujainmda@gmail.com; 3Amity Institute of Virology and Immunology, Amity University, Noida 201 313, Uttar Pradesh, India; hridayesh.prakash@gmail.com; 4Department of Food Science and Technology, University of Agriculture Science, Bangalore 560 065, Karnataka, India; ssnk4343@gmail.com; 5Lake Erie College of Osteopathic Medicine, Bradenton, FL 34211, USA; elise.sandberg@gmail.com

**Keywords:** aromatase, CYP19A1 gene, aromatase inhibitor, lncRNA, miRNA, breast cancer, ovarian cancer, prevention, therapy

## Abstract

Aromatase inhibitors (AIs) such as anastrozole, letrozole, and exemestane have shown to prevent metastasis and angiogenesis in estrogen receptor (ER)-positive breast and ovarian tumors. They function primarily by reducing estrogen production in ER-positive post-menopausal breast and ovarian cancer patients. Unfortunately, current AI-based therapies often have detrimental side-effects, along with acquired resistance, with increased cancer recurrence. Thus, there is an urgent need to identify novel AIs with fewer side effects and improved therapeutic efficacies. In this regard, we and others have recently suggested noncoding RNAs (ncRNAs), specifically microRNAs (miRNAs) and long noncoding RNAs (lncRNAs), as potential molecular targets for utilization in modulating cancer hallmarks and overcoming drug resistance in several cancers, including ER-positive breast and ovarian cancer. Herein, we describe the disruptive functions of several miRNAs and lncRNAs seen in dysregulated cancer metabolism, with a focus on the gene encoding for aromatase (CYP19A1 gene) and estrogen synthesis as a novel therapeutic approach for treating ER-positive breast and ovarian cancers. Furthermore, we discuss the oncogenic and tumor-suppressive roles of several miRNAs (oncogenic miRNAs: MIR125b, MIR155, MIR221/222, MIR128, MIR2052HG, and MIR224; tumor-suppressive miRNAs: Lethal-7f, MIR27B, MIR378, and MIR98) and an oncogenic lncRNA (MIR2052HG) in aromatase-dependent cancers via transcriptional regulation of the CYP19A1 gene. Additionally, we discuss the potential effects of dysregulated miRNAs and lncRNAs on the regulation of critical oncogenic molecules, such as signal transducer, and activator of transcription 3, β-catenin, and integrins. The overall goal of this review is to stimulate further research in this area and to facilitate the development of ncRNA-based approaches for more efficacious treatments of ER-positive breast and ovarian cancer patients, with a slight emphasis on associated treatment–delivery mechanisms.

## 1. Introduction

While breast cancer is the fourth leading cause of cancer-related mortality in women, ovarian cancer is the most aggressive and fatal of all female reproductive cancers, with a five-year mortality rate of ~80% when diagnosed in the 4th stage [1]. With such powerful rates of prevalence and mortality in these respective cancer types, researchers have been on a strong pursuit to develop novel, biomarker-driven therapeutic agents against breast and ovarian cancer. In congruency with these efforts, the Food and Drug Administration (FDA) has approved several biomarker-based precision medicines, including Trastuzumab, which is indicated for breast cancer patients demonstrating human epidermal growth factor receptor 2 (HER2) gene amplification with subsequent elevations in protein level [2,3]. Unfortunately, despite recent advancements, highly specific molecular therapeutic strategies have still yet to reach maximum efficacy in clinical settings.

Current therapeutic strategies against breast and ovarian cancer involve blocking the estrogenic effects on tumors via endocrine therapy [4]. In this regard, conventional endocrine therapeutics entail treatment of estrogen receptor α (ERα) in ER-positive female cancers. ER-modulating medications include tamoxifen and fulvestrant. Aromatase inhibitors (AIs), including letrozole, anastrozole, and exemestane, are also used as anticancer agents, for they suppress estrogen production in peripheral tissues by inhibiting the activity of the aromatase enzyme [5,6]. Aromatase (EC 1.14.14.1) is a critical enzyme that regulates the rate–limiting step involved in estrogen biogenesis, initiating the conversion of androgens to estrogens via three successive hydroxylation’s and elimination of carbon at the 19th position of the androgens [7]. Aromatase is a product of the *CYP19A1* gene, and it is an essential member of the cytochrome p450 superfamily, subfamily 19. *CYP19A1* is a 123 Kb gene located on chromosome 15 at position q21.1, consisting of ten exonic and nine intronic regions, and it is found in all body tissues. Structurally, aromatase is a dimer of two complex polypeptides—one specific to cytochrome P450, a transcriptional product of the *CYP19A1* gene, and the other to flavoprotein NADPH-cytochrome P450 reductase [8].

Overexpression of aromatase is associated with elevated systemic levels of estrogen, and is linked with the pathogenesis of hormone-associated disorders, including breast, ovarian, and endometrial cancer, as depicted in Figure 1 [9]. Recently, several studies have demonstrated the therapeutic potential of AIs used to treat estrogen-dependent cancers [10]. Unfortunately, the clinical uses of various AIs (e.g., letrozole, anastrozole, and exemestane) are limited due to detrimental side effects, such as deep vein thrombosis, cataract, loss of appetite, and osteoporosis, as well as the development of resistance to these drugs [11]. In this regard, noncoding RNAs (ncRNAs) have the potential to provide therapeutic and diagnostic alternatives for conventionally used AIs. Noncoding RNAs were once considered “junk DNA”; however, more recent studies have revealed that they perform a significant role in many critical biological functions associated with cancer, including cell proliferation, metastasis, and drug response [12]. Furthermore, ncRNAs (which include microRNAs [miRNAs] and long noncoding RNAs [lncRNAs]), have been implicated in regulating chromosome structure and DNA methylation patterns governing epigenetic regulation [13]. From a clinical perspective, ncRNAs hold promise as useful diagnostic/prognostic biomarkers due to their presence in several body fluids, including serum, saliva, and urine, suggesting that conventional invasive procedures such as tissue biopsy may not be required [14]. Additionally, they are highly specific, altered expression based on disease progression and clinicopathological characteristics [15].

In this review, we discuss the roles of ncRNAs, such as miRNAs and lncRNAS, in ER-positive breast and ovarian cancers and the potential applications of these molecules as novel targets for improved therapy.

## 2. Methodology for Literature Search and Selection

We utilized standard reporting items for systematic reviews and a meta-analysis (PRISMA) to analyze the literature for this review Figure 2 [16]. We considered the most up-to-date scientific research with no year restriction to provide a comprehensive set of information to readers. We performed a literature search employing multiple online databases dated 1/Jan/2021, such as PubMed (470), ScienceDirect (120), Willey (70), Google Scholar (630), and Lexis Nexis (523). Primary screening led to the identification of 343 studies after duplicate removal. These studies were selected based on the title, keywords, and abstract content, which were used to screen for eligibility of relevant results. After further review, we were able to narrow down 35 relevant studies. This step was followed by qualitative evaluation involving full-text analysis, exclusion of data which were incomplete, and highly qualitative data or bioinformatic studies employing data from The Cancer Genome Atlas database, producing 14 results. For post-qualitative screening, we eliminated reviews, meta-analysis, meeting reports, patents, book chapters, and conference abstracts, yielding 10 relevant articles for final analysis. The literature search was performed by an individual author (TSB) in consultation with a senior author (AJ) of this review.

## 3. Noncoding RNAs (miRNAs and lncRNA) and Their Aberrant Expression in ER-Positive Breast Cancer

Despite recent advances in cancer therapeutic and diagnostic methodologies, breast cancer remains a significant clinical, scientific, and societal challenge [17], representing the second leading cause of cancer-related mortalities globally [18]. Recent studies have shown that ~70% of breast cancers can be classified as estrogen receptor alpha-positive, belonging to the molecular subtype luminal A or luminal B [16]. Though the mechanisms involved in breast cancer etiology are not fully understood, the present scientific data demonstrate a robust correlation between estrogen levels and the development and progression of the disease [19]. The proliferative and metastatic potential of ER-positive breast cancer cells are often governed by the estrogen signalling pathway; hence, estrogen-blocking strategies for hormone-sensitive tumors have been effective in reducing tumor growth. Aromatase is an example of a target for estrogen-blocking therapeutics, as it is an enzyme involved in regulating estrogen production in visceral fat cells. Recently, miRNAs and lncRNA have been identified as potential target molecules that can downregulate levels of aromatase at the mRNA level [20,21,22]. Contrary to the conventional notion, recent work has found them to serve as important functional, regulatory molecules [23]. Therefore, a more thorough understanding of the complex ncRNA (miRNAs and lncRNA)-related networks and interactions in cells might provide a unique opportunity for the development of improved molecular suppressors against aromatase as alternatives to conventionally used AIs. In the following section, we discuss several ncRNAs that have been found to have regulatory potential in breast cancer progression.

### 3.1. Aromatase Associated with miRNAs Aberrantly Expressed in Breast Cancer

#### 3.1.1. Lethal-7f

Lethal-7f is a miRNA located on chromosome 9 at position 9q22.32 that can modulate developmental timing and cellular differentiation [24,25], and its dysregulated expression has been associated with cancer initiation and progression [26]. Lethal-7f has 13 family members, encoding 9 mature miRNAs (*let-7a*, *let-7b*, *let-7c*, *let-7d*, *let-7e*, *let-7f*, *let-7g*, *let-7i*, and *miR-98*), which have overlapping functions due to high sequence similarity [27]. 

Mechanistically, lethal-7f downregulates β2-adrenergic receptor (β2-AR) levels by interacting with the 3′-UTR (untranslated region) of the gene coding β2-AR levels in human epidermal growth factor receptor 2 (HER2)-positive breast cancer patients, as demonstrated in MCF-7, SKBR3 and BT474 cell lines. β2-adrenergic receptor disruption leads to suppression of downstream molecules, such as matrix metalloproteases (MMPs) and vascular endothelial growth factor (VEGF), which reduces tumor invasion and angiogenesis [28]. Shibahara et al. (2012) have demonstrated a negative correlation between lethal-7f and *CYP19A1* mRNA levels in breast cancer tissues (n = 3), further validated using human breast cancer cell lines, MCF-7 and SK-BR-3 (Table 1). Lethal-7f has also been shown to contribute to estrogen depletion and decreased tumor cell proliferation via sponging of the *CYP19A1* gene (2.44-fold decrease) interacting with 3′-UTR, as depicted in Figure 3 [29]. Further studies on lethal-7f are still required with a large number of patients.

Furthermore, lethal-7f has been shown to modulate tumor cells’ sensitivity to chemotherapy and radiation in various human malignancies [30]. Therefore, lethal-7f represents a potential therapeutic molecule against breast cancer.

#### 3.1.2. MIR125B1

MIR125B1 is a miRNA located on chromosome 21 at position 21q221.31 that can modulate several cellular processes [31]. Based on its cellular context, it acts as a tumor proliferator, or oncogene, in breast cancer [32], ovarian cancer [33], gliomas [33]. Furthermore, several studies have demonstrated the proliferative potential of MIR125B1in breast cancer via its targeting of the tumor suppressor *p53* gene, which upregulates proliferative tumor pathways like PI3K (Phosphatidylinositol 3-kinase) pathways. Subsequently, this promotes upregulation of several oncogenic pathways, including the Akt/mTOR axis. Akt is a protein kinase B that, when stimulated, upregulates mechanistic target of rapamycin (mTOR), which contributes to many growth processes of cells and further contributes to tumor proliferation [34,35]. Vilquin et al. (2015) have demonstrated the proliferative potential of MIR125B1 via its ability to upregulate aromatase levels in breast cancer tissues (n = 65) compared to an equivalent number of healthy controls. Mechanistically, MIR125B1 upregulated the Akt/mTOR pathway and reduced the therapeutic potential of letrozole in breast cancer patients, as depicted in Figure 3 [34]. Furthermore, the proliferative role of MIR125B1in ER-positive breast cancer was validated by Vilquin et al. (2015), who employed MCF-7 cell line subjected to an increasing concentration (1, 3 and 5 µm) of letrozole drug [34]. Based on these scientific findings, we might conclude that MIR125B1 blockers represent a potential therapeutic target against ER-positive breast cancer (Table 1). In further support of this idea, suppression of MIR125B1 has demonstrated the ability to not only reduce letrozole resistance but to suppress several other oncogenic molecules affecting tumor proliferation. Furthermore, upregulated levels of MIR125B1 can serve as a prospective diagnostic biomarker for ER-positive breast cancer, as depicted in Figure 3.

#### 3.1.3. MIR27B

MIR27B is a miRNA located on chromosome 9 at position 9q22.32 that modulates tumor suppression in breast cancer [36,37]. MIR27B is a well-validated tumor suppressor and mechanistically modulates critical oncogenic molecules such as the nuclear receptor subfamily 5 group A member 2 gene (*NR5A2*) and response element-binding protein (*CREB1*). MIR27B mRNA binds to the 3′-UTR of the *NER5A2* gene, a critical protein-coding gene involved in the expression of essential genes, such as DNA binding zinc finger transcription factor and cholesterol biosynthesis, which promote suppression of cancer proliferation. Similarly, MIR27B mRNA binds with the 3′-UTR of the *CREB1* gene, which blocks estrogen-induced transcription due to lack of inducer transcripts for DNA binding protein suppressing tumor proliferation [38].

Demonstrating the regulatory potential of MIR27B in overcoming tamoxifen-resistant ER-positive breast cancer, Zhu et al. (2016) illustrated the reduced tamoxifen-resistance potential of MIR27B mRNA inhibiting *NR5A2* and *CREB1* genes in MCF-7 and TAM-1 (tamoxifen-resistant) cell lines. Furthermore, Zhu et al. (2016) validated the reduced tamoxifen-resistance in breast cancer from 53 tumor samples, compared with 19 healthy tissue samples (Table 1) [38]. Based on these scientific findings, we might conclude that MIR27B represents a potential therapeutic target against ER-positive breast cancer. Suppression of MIR27B not only reduces tamoxifen resistance; it also suppresses several other oncogenic molecules affecting tumor proliferation, as depicted in Figure 3.

#### 3.1.4. MIR155

MIR155 is a miRNA located on chromosome 21 at position 21q21.3 that modulates several oncogenic pathways in breast cancer patients [39]. It is a well-validated tumor proliferator that is highly overexpressed in several solid tumors, such as thyroid carcinoma [48], breast cancer [41,49,50], colon cancer [41], cervical cancer [51], pancreatic ductal adenocarcinoma (PDAC) [52], and lung cancer, and the aberrant expression of MIR155 is associated with poor prognosis [53]. Mechanistically, MIR155 suppresses MIR143, a well-documented tumor suppressor [54], via targeting the activation of STAT3 gene targeting c/EBPB (a transcriptional activator for MIR143) [40]. 

Demonstrating the regulatory potential of MIR155 in overcoming letrozole-resistance in ER-positive breast cancer, Bacci et al. (2016) conducted in-vivo experiments using MCF-7 and ZR75-1 (letrozole-resistant) breast cancer cell lines. Additionally, Bacci et al. (2016) validated the in vitro experiments by performing in vivo experiments using female Ncr foxhed nude mice 6 to 8 weeks old, who were given 1 mg/kg of letrozole for 21 days (Table 1). Furthermore, the authors further demonstrate an upregulated expression of MIR155 in letrozole-resistance in ER-positive breast cancer cells. Mechanistically, MIR155 upregulates *SLC16A3* gene encoding monocarboxylate transporter 4 protein (MCT4) and glucose transporter 1 (*GLUT-1*) gene, increasing letrozole resistance in ER-positive breast cancer [40]. Therefore, using a suitable MIR155 miRNA blocker molecule might act as a potential therapeutic molecule against letrozole resistance in ER-positive breast cancer. Furthermore, upregulated levels of MIR155 can serve as a prospective diagnostic biomarker against ER-positive breast cancer, as depicted in Figure 4.

#### 3.1.5. MIR221 and MIR222

MIR221 and MIR222 are two highly homologous microRNAs located on chromosome X at position Xp11.3, whose upregulation has been recently demonstrated in several types of human tumors [41]. Overexpression of MIR221 & MIR222 is associated with several advanced malignancies via suppression of critical cell cycle regulators, such as p27Kip1 [55]. The inverse correlation amongst p27Kip1 with MIR221 & MIR222 was confirmed in glioblastomas, thyroid papillary carcinomas, breast cancer, hepatocellular carcinoma, and lung cancer [56]. Mechanically, MIR221 & MIR222 upregulation in breast cancer was correlated with downregulated *p53* gene profile, leading to an upregulated expression of critical oncogenic pathways such as the PI3K (Phosphatidylinositol 3-kinase) pathways, and subsequent upregulation of several oncogenic pathways such as the Akt (protein kinase B) and mTOR pathway, further contributing to tumor proliferation [34]. Demonstrating the regulatory potential of MIR221 & MIR222 in overcoming fulvestrant resistance in ER-positive breast cancer, Rao et al. (2011) conducted in vivo experiments using MCF-7 (fulvestrant-resistant) breast cancer cell lines dosed with 100 nM of fulvestrant (Table 1). Mechanistically, MIR221 & MIR222 upregulated β-catenin mRNA levels, contributing to increased fulvestrant resistance in MCF-7 breast cancer cell lines [57]. Furthermore, β-catenin is a critical oncogenic molecule associated with crucial roles; for instance, promoting cell-cell adhesion [58], upregulating cell proliferation, mediating EMT, and enhancing cell resistance to chemoradiotherapy in breast cancer [42]. Therefore, using a suitable *MIR-221* & *MIR-222* blocker molecule might act as a potential therapeutic agent against fulvestrant resistance in ER-positive breast cancer. Furthermore, upregulated *MIR-221* & *MIR-222* can serve as a prospective diagnostic biomarker against ER-positive breast cancer, as depicted in Figure 4.

#### 3.1.6. MIR128-1

MIR128-1 is an endogenous miRNA that is 18–24 nucleotides long and located on chromosome 2 at position 2q21.3. It has been shown to modulate tumorigenesis and metastasis in several cancers. Subsequently, MIR128-1 is considered a key biomarker for the diagnosis and prognosis of cancers, as well as an effective agent for targeted therapy of malignant tumors. Furthermore, MIR128-1 targets several oncogenes governing tumor proliferation, differentiation, apoptosis, invasion, and metastasis, providing essential cues for the development of novel therapeutic strategies for cancer prevention and treatment [43]. Mechanistically, MIR128-1 downregulates *ARPP21* (CAMP-regulated phosphoprotein 21) gene, a crucial gene that inhibits apoptosis and confers increased chemo-radiotherapy resistance in breast cancer cells [59].

Demonstrating the regulatory potential of MIR128-1 in overcoming letrozole resistance in ER-positive breast cancer, Masri et al. (2012) conducted an in vivo experiment using MCF-7 breast cancer cell lines overexpressing the aromatase gene (Table 1). Mechanistically, MIR128-1 upregulates the transforming growth factor-β receptor 1 (*TGFβR1*) gene associated with re-sensitization of letrozole resistance in ER-positive breast cancer [44]. Furthermore, TGFβR1 recruits and phosphorylates receptor-regulated Smads (R-Smads) when phosphorylated. Additionally, the *TGFβR1* gene regulates cell growth, apoptosis, differentiation, and fibrosis [60]. Therefore, using a suitable MIR128-1 blocker molecule might act as a potential therapeutic molecule against fulvestrant resistance in ER-positive breast cancer. Furthermore, upregulated MIR128-1 can serve as a prospective diagnostic biomarker against ER-positive breast cancer, as depicted in Figure 4.

### 3.2. Aromatase Associated with miRNAs Aberrantly Expressed in Breast Cancer

Long noncoding RNAs (lncRNAs) are a subtype of ncRNAs that are >200 nucleotides in length [13,14,61,62,63], and several have been implicated in the development and progression of breast cancer [31]. LncRNAs can act as promoters or suppressors during breast cancer progression via molecular mechanisms that include modulating cellular proliferation, invasion, apoptosis, and drug resistance [12,14]. Unlike miRNAs, knowledge of the functional roles of lncRNAs is still somewhat limited [64]. In the subsequent section, we discuss the potential of several lncRNAs in regulating estrogen levels via the regulation of the aromatase enzyme.

#### MIR2052HG

*MIR2052HG* is a lncRNA that is 32.4 kb in length and located on chromosome 18 at position 8q21.11. *MIR2052HG* has been associated with breast cancer [45,46]; however, the mechanism(s) involved are not yet fully defined. A correlation has been identified between *MIR2052HG* and aromatase levels in MCF7 and CAMA-1 (ERα-overexpressing breast adenocarcinoma) cell lines. Further, *MIR2052HG* was found to regulate ERα expression through transcriptional regulation of the estrogen receptor 1 (ESR1) and ER protein degradation, as schematically outlined in Figure 5. *MIR2052HG* has been found to regulate Lemur tyrosine kinase 3 (LMTK3) transcription, in which the transcribed *MIR2052HG* and *LMTK3* gene transcripts interact with the early growth response 1 (EGR1) protein to initiate movement of the complex toward the LMTK3 locus. This results in the upregulation of aromatase, mitogen-activated protein kinase (MAPK), and protein kinase B/forkhead box O3 (Akt/FOXO3), leading to ERα and ESR1 stability, as depicted in Figure 5. The introduction of *MIR2052HG* lncRNA blockers in MCF7 and CAMA-1 cell lines led to a significant decrease in ERα [47]. Similarly, Ingle et al. (2017) demonstrated that depletion of *MIR2052HG* via siRNA resulted in decreased expression of ERα, both at the mRNA and the protein levels, in MCF7 and AC1 human gestational choriocarcinoma cell line [20]. Consistent with these results, Ingle et al. (2017) also found increased levels of *MIR2052HG* in 253 breast cancer patients, compared to 4406 healthy controls [21]. 

*MIR2052HG* was found to regulate Akt-dependent FOXO3 expression. Its depletion led to overexpression of FOXO3 in MCF7/AC1 cells, resulting in a significant decrease in ERα mRNA and protein levels (Table 1). Therefore, based on the results discussed above, the depletion of *MIR2052HG* may be associated with reduced aromatase levels, decreased estrogen levels, and reduced cellular proliferation and tumor metastasis [21]. However, further studies are warranted to better understand the involved mechanisms and pathways modulated by *MIR2052HG*.

## 4. MiRNAs and Their Aberrant Expression in ER-Positive Ovarian Cancer

Universally acknowledged as a “silent killer,” ovarian cancer is often not diagnosed until it has progressed to advanced stages. This is in part due to a delayed onset of symptoms and lack of efficient screening [65,66]. In support of this, statistics have shown that >70% of ovarian cancer cases are not diagnosed until the tumor has progressed to an advanced stage, in which the average five-year survival rate is only ~50% [67]. Epidemiological data suggests that ovarian cancer induction and proliferation are impacted by lifetime estrogen exposure [68]. Experiments have revealed that ovarian cancer shares estrogen-regulated pathways with breast, cervical, and endometrial cancers [69], as the estrogens stimulate tumor growth by binding to and activating the ER. Several large-scale studies have displayed that ~36% of ovarian cancers are ER-positive [70]. Therefore, estrogen-blocking strategies have high applicability in regulating estrogen production in the visceral fat cells [71].

According to the National Comprehensive Cancer Network (NCCN) guidelines, hormonal therapies including anastrozole (aromatase inhibitor), letrozole (aromatase inhibitor), leuprorelin acetate (gonadotropin-releasing hormone analog), megestrol acetate (synthetic progestin), and tamoxifen (anti-estrogen) are all classified as approved treatments for recurrent forms of epithelial ovarian cancers [72]. Therefore, the development of additional hormone-based therapies may be beneficial for these patients. In this regard, miRNA provides a lower dose inhibitory analogy, providing easier manipulation due to absolute expression levels and avoiding oligonucleotide-associated toxicity [73]. In the following section, we discuss the roles of several miRNAs that have been implicated in ovarian cancer progression.

### 4.1. MIR224

MIR224 is located on chromosome X at position Xq28. MIR224 performs critical gene-regulating functions involved in chemoresistance in A2780CP/A2780S and C13/OV2008 ovarian cancer cells, via regulating the PRKCD (protein kinase C delta) pathway. Additionally, the oncogenic potential of MIR224 was validated using 41 ovarian papillary serous carcinomas (OPSC) [74]. Hu et al. (2016) demonstrated the proliferative potential of MIR224 via targeting the KLLN pathway in HO8910 (low metastatic ability) and HO8910PM (high metastatic ability) ovarian cancer cell lines. Hu et al. (2016) further illustrated KLLN protein’s (Killin, P53 Regulated DNA Replication Inhibitor) suppression via MIR224, which fostered a downstream target, cyclin A, to initiate proliferation in epithelial ovarian cancer cells [74]. Yao et al. (2010) demonstrated a strong correlation amongst upregulated expression levels of *CYP19A1* mRNA (5-fold upregulation) and MiR-224 in KGN cell lines. Mechanistically, MIR224 enhanced TGF-β1protein levels, which induced GC proliferation by targeting Smad4 protein and led to a subsequent increase in the *CYP19A1* mRNA levels. Furthermore, Yao et al. (2010) validated the proliferative potential of MIR224 by employing adult ICR female mice Table 2 [74]. Similarly, Lite et al. (2019) illustrated the aromatase-enhancing potential of MIR224 using three-months-old nulliparous rats [75].

Based on these results, we propose that MIR224 represents a potential biomarker for ovarian cancer (Figure 6). However, further studies are warranted, particularly those designed to delineate the underlying molecular mechanisms involved in MIR224-associated ovarian cancers. Unfortunately, additional human studies are warranted to mandate the diagnostic potential of *MIR-378*.

### 4.2. MIR378

MIR378, annotated as MIR-378, MIR 378, miRNA378, hsa-mir-378, and hsa-mir-378a, is located on chromosome 5q32 [76,77] and its overexpression has been correlated with inhibited cell proliferation and induced apoptosis [80]. Mechanistically, MIR378 inhibits ERRγ and GA-binding protein-α at the mRNA level. This suppression upregulates a critical tumor suppressive gene, PGC-1 β, which controls oxidative metabolism. Xu et al. (2011) demonstrated that decreased levels of MIR378 in porcine ovaries correlated with increased aromatase and estrogen levels. Mechanistically, MIR378 suppressed aromatase (3-fold downregulation) by targeting specific sites within the aromatase 3′-UTR (Table 2) [76]. A similar study performed by Pan et al. (2015) demonstrated that upregulation of MIR378 in cumulus cells of porcine ovaries resulted in decreased expression of prostaglandin-endoperoxide 193 synthase 2 (PTGS2) and hyaluronan synthase 2 (HAS2) transcripts, both of which are associated with anti-angiogenesis. Furthermore, overexpressed MIR378 was found to downregulate aromatase (2.75-fold downregulation) by targeting its 3′-UTR [81] Table 2. Therefore, MIR378 might have potential as a therapeutic target against ovarian cancer; not only acting as an AI but also by targeting pathways associated with the suppression of tumor progression and proliferation (Figure 6).

### 4.3. MIR98

MIR98 are short (20-24 nucleotide) noncoding RNAs located on chromosome X at Xp11.22. MIR98 is a well-documented tumor suppressor, having demonstrated the ability to inhibit cell survival, cell proliferation, tumor growth, and invasion by targeting activin receptor type-1B (*ALK4*) and *MMP11*. Mechanistically, the tumor suppressor potential of MIR98 was validated using the luciferase reporter assay, where MIR98 was found to bind to the 3′-untranslated region of *ALK4* and *MMP11* mRNA. MMP11 accelerated the degradation of serine protease inhibitor α1-antitrypsin and insulin-like growth factor binding protein-1 (IGFBP-1), induced differentiation of a desmoplastic reaction surrounding the cancer stroma through cleavage of collagen VI and stimulated tumor progression and metastasis [78]. Similarly, ALK4 (activin receptor type-1B) activated cellular proliferation by targeting the Smad4 protein [82]. Panda et al. (2012) demonstrated the tumor-suppressive potential of MIR98 via targeting of *CYP19A1* mRNA, which demonstrated reduced estrogen levels in 52 patients compared to 15 healthy individuals. Furthermore, MIR98 reduced the expression of *CYP19A1* mRNA (3.5-fold downregulation). Panda et al. (2012) validated the *CYP19A1* regulatory potential of MIR98 deploying in vitro (Ishikawa cells) and in vivo (52 patients and 15 healthy control samples) experimentation (Table 2). Additionally, Panda et al. (2012) correlated the inhibitory potential of MIR98 with clinicopathological features, including age, tumor grade, tumor size, and staging. Furthermore, the sponging of the *CYP19A1* mRNA by MIR98 contributed to estrogen depletion and decreased tumor cell proliferation, migration, invasion, and angiogenesis, suggesting application of MIR98 as a therapeutic and diagnostic molecule in clinical settings [79].

## 5. Mechanisms of Various Delivery Methods for Therapeutic miRNAs and Future Direction for Efficacious Use in the Clinical Setting

Over the past few decades, our approach toward cancer therapeutics has transformed and continues to transition, from focusing on treating invasive cancers towards placing a greater emphasis on early detection and prevention. Additionally, advances in targeted therapies have been made to reduce detrimental side effects in patients [83,84]. For example, targeted inhibitors of aromatase have been developed to regulate estrogen synthesis in cancer cells [85]. Inhibition of such rate–limiting biomolecules may be a key to long-term treatment of a large subset of the population, while also having limited adverse side effects. However, treatment with conventional AIs, such as letrozole, anastrozole, and exemestane, often results in detrimental side effects such as osteoporosis, carpal tunnel syndrome, changes in appetite, constipation, diarrhoea, and vaginal bleeding. Thus, the developments of novel AIs with reduced side effects are needed. In this regard, ncRNA-based approaches to target aromatase may significantly improve conventional inhibitors, as outlined in Figure 3, Figure 4, Figure 5 and Figure 6.

Within our comprehensive search of the literature, we discussed one dysregulated lncRNA and six miRNAs that have been associated with breast cancer. Similarly, we examined three dysregulated miRNAs related to ovarian cancer. Let-7f miRNA has been reported as the most potent inhibitor of aromatase, leading to a significant decrease in aromatase levels in MCF-7 and SK-BR-3 breast cancer cell lines. Additionally, these results were validated using three breast tissue samples, suggesting that let-7f may provide a novel target to substitute for conventionally-used AIs in the treatment of estrogen-sensitive breast cancers. Further studies are warranted to validate the therapeutic potential of let-7f as a substitute to conventional AI therapy [29]. Likewise, the evaluation of several ncRNAs associated with ovarian cancer demonstrated that MiR98 miRNA might be the most promising inhibitor of aromatase, as targeting this lncRNA for depletion led to a significant decrease (3.5-fold downregulation) in aromatase levels in the endometrial tumor cell line (Ishikawa cells). Of clinical relevance, similar results were found in 52 patient samples, when compared to 15 healthy control samples [79]. Thus, it might be possible to use ncRNAs as potential biomarkers and to include them in biomarker panels for diagnostic and prognostic applications in cancer patients. Furthermore, critical shortcomings associated with ncRNA delivery, immunogenicity, and suppression of oncogenic ncRNA have been well studied, and their inferences have been discussed in the subsequent section.

### 5.1. Approaches for Systemic Delivery of Therapeutic ncRNAs

ncRNA, such as microRNAs and long noncoding RNAs, are critical regulatory molecules dysregulated in cancer in tissue and stage-specific manner. Cellular penetration of ncRNA requires improved stabilization against degradation, requiring the employment of optimized chemical structures. Being structurally identical to siRNAs, similar pharmaceutical formulations encompassing lipid-based delivery vehicles, polymeric nanoparticles, and viral systems can be utilized to increase stability and enhance the pharmacokinetic potential of the associated oligonucleotides [86,87,88].

Furthermore, Sharma et al. (2017) exhibited enhanced efficacy of anti-miR-191, which was achieved using MCF-7 and ZR-75-1 human breast cancer epithelial cell lines. Conjugated liposomes with anti-miR-191 reported enhanced apoptosis and suppressed metastasis [89]. Despite the ease of manufacturing and protection against nucleotide degradation, the liposomal delivery mechanism is plagued with hurdles, including scalability, reliability, chemical instability, and denaturation [90]. 

Similarly, structural analogs, called exosomes, can also be used for the transfer of miRNA amongst cells in vivo. Exosomes are composed of the phospholipid bilayer, and the biogenesis of exosomes ensures increased biocompatibility with minimal toxicity. Recently, Gai et al. (2020) illustrated lncRNA exosomal encapsulation of metallothionein 1D pseudogene lncRNA (MTDP-lncRNA) in H1299 non-small cell lung cancer cells, which generated a favourable therapeutic effect in tumor cells via regulation of the miR-365a-3p/NRF2 axis [91]. Unfortunately, exosome manufacturing faces complications within pharmaceutical development, production, high immunogenicity, and potential biological impurities.

Furthermore, to overcome the problems associated with the aforementioned conventionally used processes, several polymers, such as polyethyleneimine (PEI), polyurethane-polyethylene imine copolymer (PU-PEI), poly lactic-co-glycolic acid (PLGA), and silica-based polymers are used as possible mechanisms for oligonucleotide delivery [92]. Comparative to the liposomal delivery mechanism, polymer-based delivery mechanisms have advantages, such as controlled delivery via regulation of the degree of cross-linking of the lattice [93]. Peng et al. (2017) demonstrated the efficient transfection of miR-200c mimics in U2OS human osteosarcoma cell lines using AMD3465 (P-SS-AMD) polymer conjugates, which were synthesized using HPMA (N-(2Hydroxypropyl) methacrylamide) and methacrylamide monomer conjugated with AMD3465, via a self-immolative disulfide linker. Transfection with miR-200c mimics demonstrated improved penetration, reduced degradation, and inhibition of metastasis in U2OS human osteosarcoma cell lines [94,95]. Unfortunately, polymeric systems suffer severe toxicity and unpredictability due to unwanted interactions with encapsulated oligonucleotides and biological proteins [96].

Additionally, bacteriophages have also been recently employed to develop virus-like particles, which have been used to transfer oligonucleotides and drug molecules. This has been performed with varied success. Wang et al. (2016) demonstrated the inhibitory potential of the MS2 bacteriophage virus-like particle (VLP)-based microRNA delivery system, cross-linked with the HIV TAT peptide in conjugation with miR-122, used to inhibit metastasis and tumor progression in Hep3B, HepG2, and Huh7 hepatocellular carcinoma (HCC) cell lines and Hep3B-related animal models. Unfortunately, similar to exosomes, concerns regarding high immunogenicity limit their applicability from a clinical perspective.

Depending upon the size and nature of the ncRNA discussed in the study, any of the technologies, as mentioned earlier, may be employed for the effective transfer of potential therapeutic molecules in vivo. Unfortunately, detailed reviews are warranted to prove clinical applicability.

### 5.2. Suppression of Oncogenic ncRNAs

Due to the universal availability of lncRNAs in body fluids, lncRNAs are promising therapeutic targets for cancer therapy. Their complex dynamic structure, coupled with their specific expression, provides a possible target for biomarker-driven, personalized medicine for cancer treatment. During our analysis, distinguishing lncRNA regulatory elements and upregulated patterns were observed, highlighting the need for technologies that perform functional analysis and target epigenetic modifications. Antisense oligonucleotide (ASO) is one such technology composed of a native or chemically modified (phosphorothioated) single-stranded antisense oligonucleotide, 6mer long at the central part and with RNA nucleotide at the flanking regions. Gong et al. (2018) demonstrated the inhibitory potential of *MALAT1*-specific ASO and nucleus-targeting TAT peptide co-functionalized Au nanoparticles (ASO-Au-TAT NPs). ASO-Au-TAT NPs enclosed *MALAT1*, which enhanced nuclear internalization and exhibited excellent biocompatibility in A549 lung cancer. Furthermore, reduced MALAT1 expression led to reduced metastatic tumor nodule formation in vivo [94].

Similarly, structural analogs, called locked nucleic acid GapmeRs (LNA GapmeRs), are 16 nucleotides long and contain chemically modified LNA (locked nucleic acid) in flanking arms, whereas the gap section lacks LNAs. Salehi et al. (2019) exhibited the transfection of *PVT1* antisense LNA GapmeRs, which decreased the viability of AEL (acute erythroid leukaemia) cells. Additionally, *PVT1* antisense LNA GapmeRs-transfected cells induced apoptosis and necrosis via downregulation of *C-MYC* protein [97]. Additionally, an antagonist to NATs (natural antisense transcripts) is an alternative to conventional lncRNA blockers, involving nucleotide fragments coded from the opposite strand of the host gene loci. Jadaliha et al. (2018) demonstrated positive regulation of expression and activity of the NAT-conjugated *PDCD4-AS1* in the MCF10A triple-negative breast cancer mammary epithelial cells line. Furthermore, NAT-conjugated *PDCD4-AS1*-conjugated lncRNA was associated with the regulation of post-transcriptional gene expression involving suppression of essential oncogenic and tumor suppressor genes [98].

Mixmer molecule is another such regulatory molecule composed of chemically modified nucleotides, such as LNAs, associated with a variety of monomers. Mixmer molecules are composed of unusual sequential nucleotides, making them immune to RNase H1 degradation. These atypical nucleotides sterically inhibit the linkages amongst lncRNA and their downstream targets, preventing epigenetic remodeling, altering gene expression, and directing alternative splicing [99]. Ivanova et al. (2007) demonstrated the applicative potential of 2’-O-methyl (OMe) oligonucleotide mixmers conjugated with locked nucleic acid (LNA) residues, as they worked as a powerful steric block in inhibiting Tat-dependent transactivation in a HeLa cell reporter system inhibiting β-galactosidase. Furthermore, the aforementioned steric blocking inhibited the replicative potential of the human immunodeficiency virus (HIV) [100].

Complementing the strategies mentioned earlier, small interfering RNAs (siRNAs) are one of the most widely acknowledged knockdown strategies, composed of double-stranded RNA which unwind into single-stranded RNA and subsequently initiate RNA-induced silencing. Prensner et al. (2014) demonstrated siRNA-based knockdown of the second chromosome on locus-associated with prostate-1 (SChLAP1), resulting in reduced cell invasion and metastasis. LncRNA SChLAP1 caused aggressive prostate cancer by preventing the tumor-suppressive activity of the SWItch/Sucrose non-fermentable (SWI/SNF) complex [101].

Furthermore, several other techniques, including deoxyribozymes and ribozymes, zinc finger nuclease (ZNF), transcription activator-like effector nuclease (TALEN), cluster regularly interspaced short palindromic repeats, nanobodies, aptamers, RNA decoys, and more, can be used as potentially regulatory technologies. Unfortunately, a detailed analysis of all listed techniques is beyond the scope of this manuscript, and will require additional inputs. Additionally, the applicability of the strategies mentioned above depends on the nature of the target molecule, just as how siRNA is not as effective of an immunogen as ASOs are. Similarly, siRNA demonstrates a high binding affinity in targeting ncRNA localized in the cytoplasm. Contrary to the above, ASOs are preferred over siRNA when targeting ncRNA localized in the nucleus. Therefore, localization, structural properties, and nature of the target molecule need to be taken into consideration while deciding a suitable gene-silencing technology for an effective cessation of oncogenic ncRNA.

The advent of powerful detection technologies, such as high-throughput sequencing, has made the deployment of novel miRNAs/lncRNA in the clinical settings quite possible. Despite these advances, there is a considerable knowledge gap in this field; for example, several of the ncRNAs discussed were validated using either human cell lines in vitro or animal models. Therefore, there is an urgent need to verify the results in multiple in vivo human studies.

## 6. Conclusions

Based on our search, we were able to demonstrate a clear correlation amongst aberrant miRNA and lncRNA expression in association with aromatase-dependent breast and ovarian cancers. Furthermore, we compiled evidence that demonstrates a direct correlation amongst several miRNAs and the *CYP19A1* gene (aromatase encoding gene). Additionally, we validated the critical role of several miRNAs in overcoming aromatase inhibitor resistance in ER-positive breast and ovarian cancer. Unfortunately, there are still gaps in understanding of the underlying mechanisms associated with aromatase inhibitor resistance in ER-positive breast and ovarian cancer. Therefore, more research is required for future developments in this area.

## Figures and Tables

**Figure 1 ijms-22-04072-f001:**
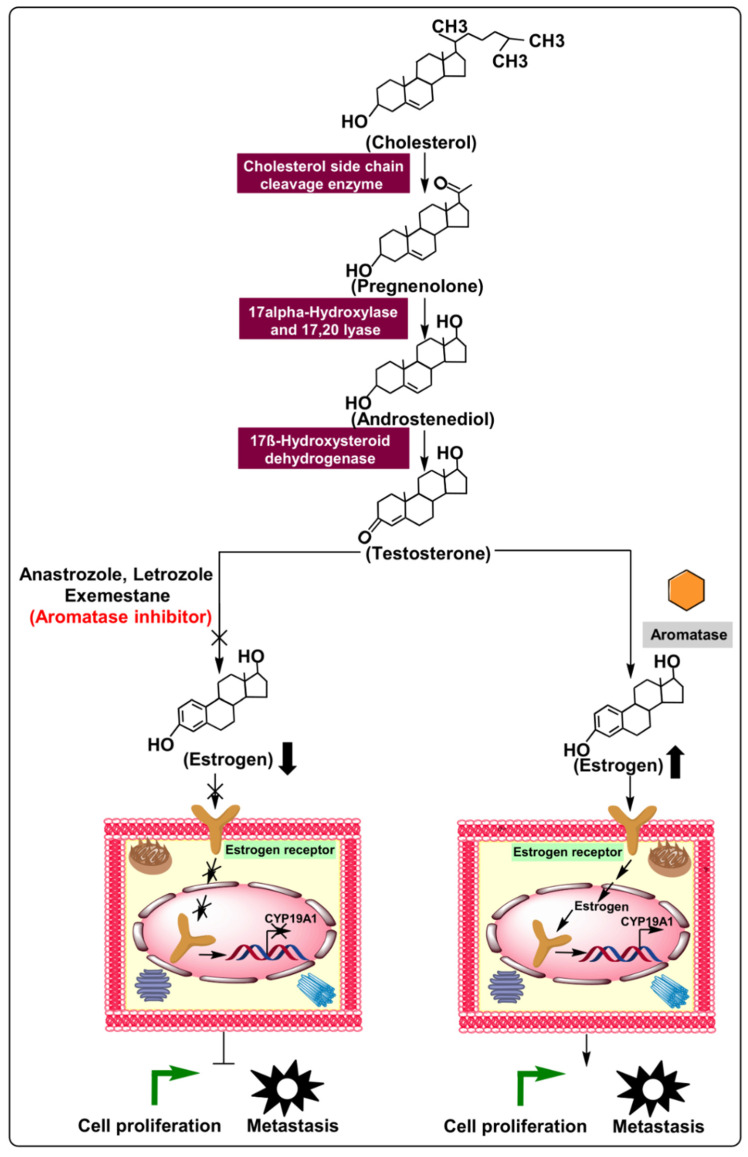
Biosynthesis of estrogen and mechanism of action of aromatase in cellular proliferation and metastasis via regulation of the estrogen production pathway. Additionally, aromatase inhibitors’ relative preferences over typical ER-positive cancerous cells demonstrate enhanced cellular proliferation and metastasis.

**Figure 2 ijms-22-04072-f002:**
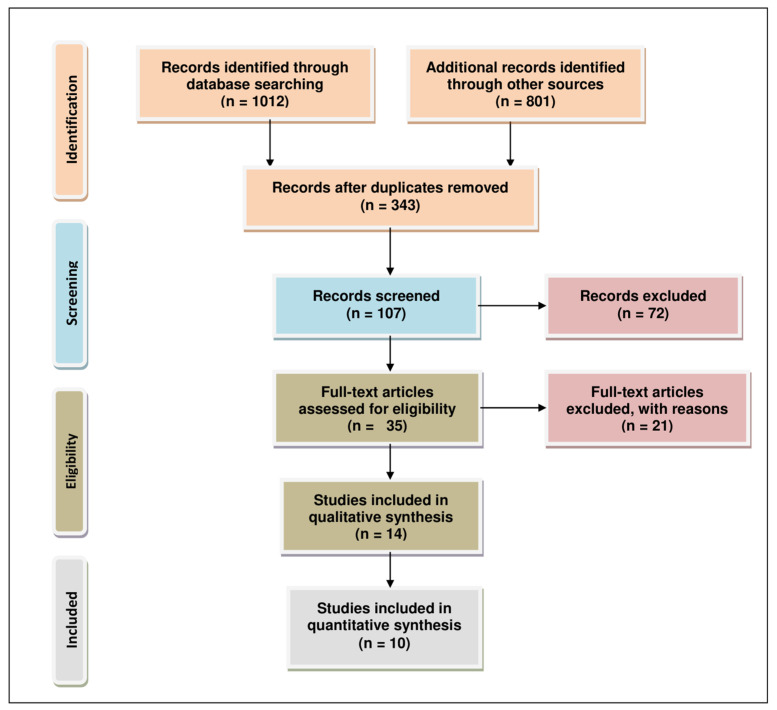
PRISMA flow chart describing the process of literature search and study selection related miRNA, lncRNA, aromatase inhibitor, and ER-positive breast and ovarian cancers. The total number of 10 relevant articles are included in this review.

**Figure 3 ijms-22-04072-f003:**
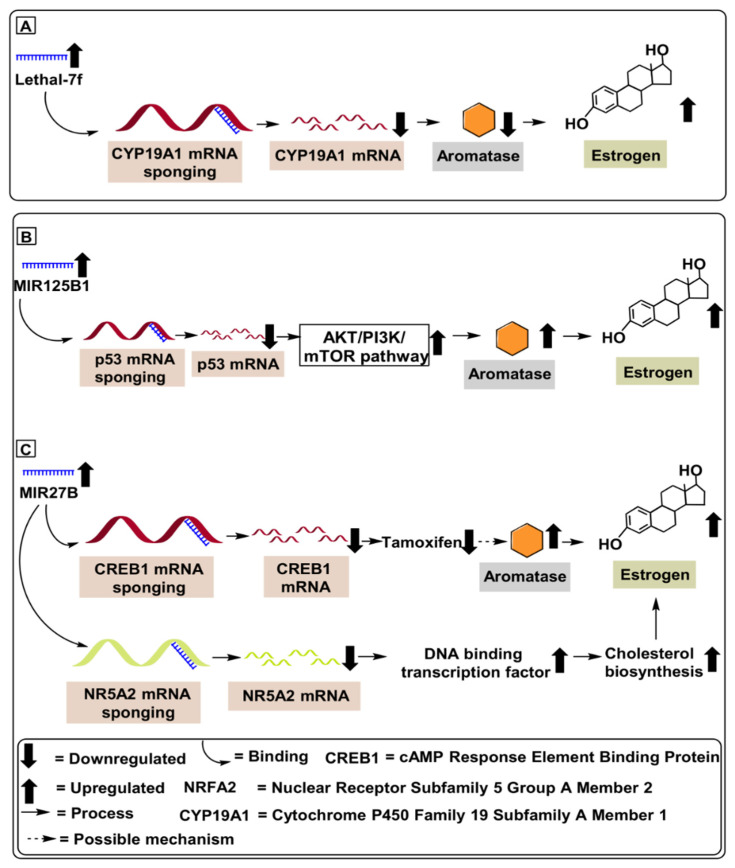
MicroRNAs involved in aromatase regulation in breast cancer. (**A**) Role of Lethal-7f downregulating *CYP19A1* mRNA levels in breast cancer patients. (**B**) Role of MIR125B1 in upregulating aromatase level by activating Akt/PI3K/mTOR pathway, contributing to chemoresistance and increasing tumor proliferation. (**C**) Role of MIR27B in downregulating *CREB1* and *NR5A2* mRNA, upregulating aromatase level and cholesterol biosynthesis in breast cancer patients.

**Figure 4 ijms-22-04072-f004:**
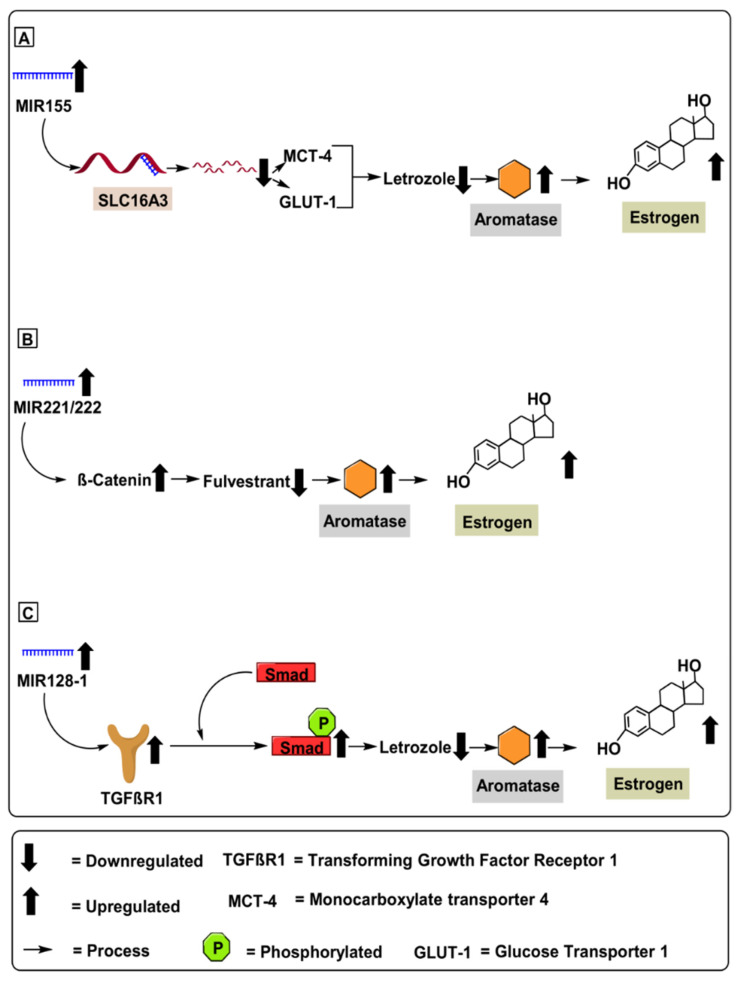
MicroRNAs involved in aromatase regulation in breast cancer. (**A**) Role of MIR155 upregulating aromatase levels via mRNA degradation of SLC16A3 gene encoding monocarboxylate transporter 4 protein (MCT4) and glucose transporter 1 (*GLUT-1*) gene increasing letrozole resistance in ER-positive breast cancer. (**B**) Role of MIR221/222 in upregulating aromatase level by activating β-catenin pathway contributing to increased fluvestrate resistance in breast cancer cells. (**C**) Role of MIR128-1 in upregulating aromatase level by activating transforming growth factor receptor-1 (TGFβ-R1) pathway, which initiates cell growth, apoptosis, differentiation and fibrosis.

**Figure 5 ijms-22-04072-f005:**
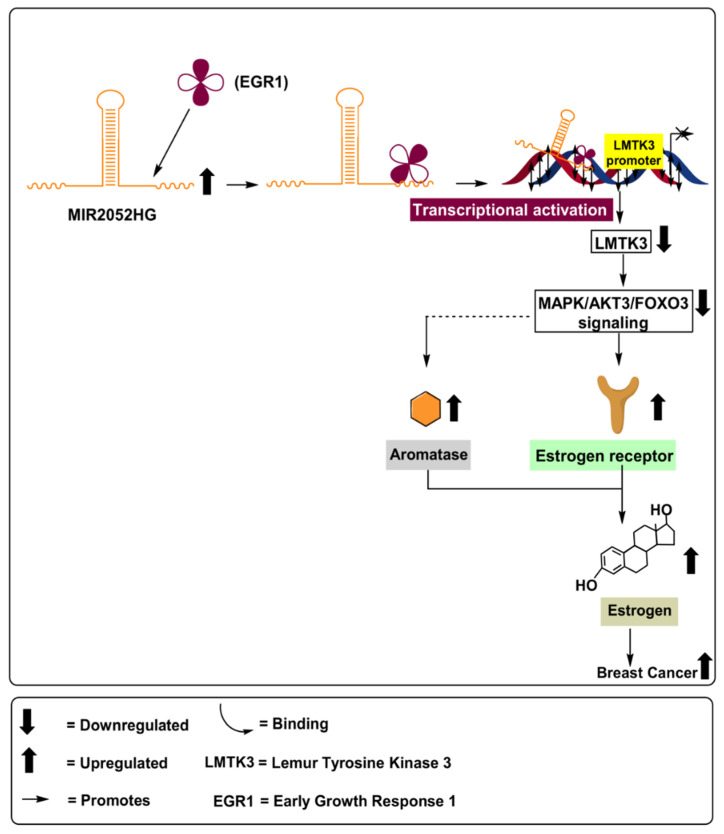
Role of lncRNA MIR2052HG in upregulating aromatase levels via activation of MAPK/Akt3/FOXo3 signalling pathways, leading to increased ERα and ESR1 stability in breast cancer patients.

**Figure 6 ijms-22-04072-f006:**
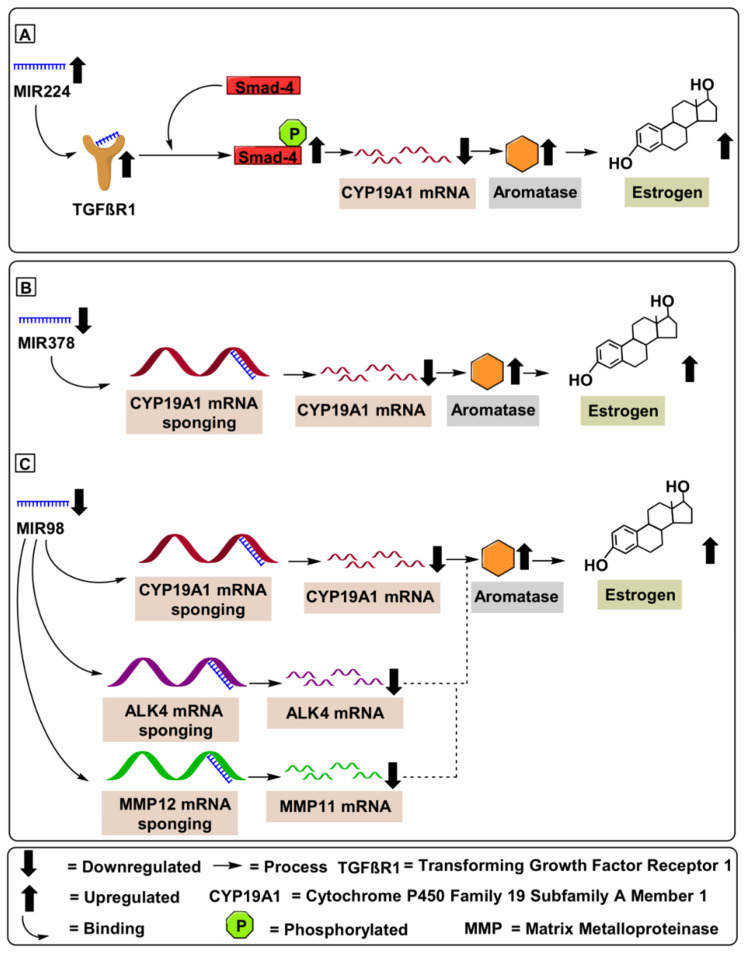
MicroRNAs involved in aromatase regulation in ovarian cancer. (**A**) Oncogenic expression of MIR224 enhances TGF-β1protein levels, inducing GC proliferation by targeting Smad4 protein, leading to a subsequent increase in the *CYP19A1* mRNA levels. (**B**) Tumor suppressive role of MIR378 downregulates *CYP19A1* mRNA levels in ovarian cancer patients. (**C**) Tumor suppressor role of MIR98 downregulates *CYP19A1* mRNA levels in ovarian cancer patients furthermore, MIR98 was found to bind to the 3′-untranslated region of *ALK4* and *MMP11* mRNA induced differentiation and cell proliferation.

**Table 1 ijms-22-04072-t001:** Dysregulated ncRNAs associated with aromatase dysregulation in breast cancer.

S. No.	Dysregulated miRNA/lncRNA	Chromosomal Location	Patient Sample	Cell Lines Used	Animal Model Used	References
Tumor-suppressive						
1	Lethal-7f	9q22.32	3 Frozen human breast cancer tissues	MCF-7 & SK-BR-3	N.A.	[24,25,29]
Oncogenic						
2	MIR125B1	21q221.31	65 Human breast cancer tissues and equivalent number of control samples	MCF-7 (Dosing: 1, 3 and 5 µm letrozole)	N.A.	[31,34]
3	MIR27B	9q22.32	53 Human breast cancer tissues and 19 healthy tissues	MCF-7 & TAM-1	N.A.	[36,37,38]
4	MIR155	21q21.3	N.A.	MCF-7 & ZR75-1	Ncr foxhed nude mice 6 to 8 weeks old given a 1 mg/kg of letrozole for 21 days	[39,40]
5	MIR221/222	Xp11.3	N.A.	MCF-7 (Dosing: 100 nM fulvestrant)	N.A.	[41,42]
6	MIR128-1	2q21.3	N.A.	MCF-7	N.A.	[43,44]
7	*MIR2052HG*	8q21.11	5221 Breast cancer blood sample	N.A.	N.A.	[45,46,47]

N.A., not available.

**Table 2 ijms-22-04072-t002:** Dysregulated miRNAs associated with aromatase dysregulation in ovarian cancer.

S. No.	Dysregulated miRNA/lncRNA	Chromosomal Location	Patient Samples	Cell Lines Used	Animal Model Used	References
Oncogenic						
1	MIR224	Xq28	N.A.	KGN cell lines	Three-months-old nulliparous rats	[74,75]
Tumor Suppressive						
2	MIR378	5q32	N.A.	KGN cell lines	Porcine ovaries	[76,77]
3	MIR98	Xp11.22	Tissues from 52 ovarian cancer patients and 15 healthy control samples	Ishikawa cells	N.A.	[78,79]

N.A., not available.

## Data Availability

Not applicable.

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
