# Peer review of "MicroRNAs and Long Noncoding RNAs as Novel Therapeutic Targets in Estrogen Receptor-Positive Breast and Ovarian Cancers"

_ijms, 2021, doi:10.3390/ijms22084072_

Round 1

Reviewer 1 Report

In the review “MicroRNAs and Long non-coding RNAs as Novel Therapeutic Targets in Estrogen Receptor-Positive Breast and Ovarian Cancers.”, Barwal et al. systematically analyze the literature on the role of aromatase inhibitors based on different RNA species in the context of estrogen receptor-positive breast cancer and ovarian cancer.

The manuscript is very informative and brings the role of RNAs more into the focus of the general discussion.

The manuscript is well organized and written. I have only minor points.

1) Fig. 1/3/4/5/6: The presentation style of the chemical structures of the cholesterol-derivates is a bit unusual, because the structure is rotated. I understand that this might have been done in order to save space. However, it would look more familiar when the chemical structures are not turned and would be presented in the regular way. 

2) Fig. 1: This figure legend would benefit from one or more sentences of explanation.

3) Conclusion: The authors have put together an interesting and informative review that could focus the attention of the readers more to the topic of RNAs in cancers. However, in the final sentences the authors may go a bit too far in terms of their wording. A review manuscript should not be called “in-depth research”, as it does not present original data. Moreover, the authors did not “validate” the correlation between specific RNAs and cetain types of cancers, because they did not validate this experimentally but only discussed published data. Please rephrase this part.

Author Response

The authors of this manuscript express their sincere thanks to the reviewer for the critical assessment of this work. The authors have acted upon the recommendations of the reviewer which have resulted in a significant enhancement in the quality of this manuscript. All modifications incorporated in the manuscript are highlighted in red color font. A “point-by-point” response to each and every comment is outlined below.

General comments:

In the review “MicroRNAs and Long non-coding RNAs as Novel Therapeutic Targets in Estrogen Receptor-Positive Breast and Ovarian Cancers.”, Barwal et al. systematically analyze the literature on the role of aromatase inhibitors based on different RNA species in the context of estrogen receptor-positive breast cancer and ovarian cancer. The manuscript is very informative and brings the role of RNAs more into the focus of the general discussion.

The manuscript is well organized and written. I have only minor points.

Response:

We are thankful to the reviewer for his/her interest and critical evaluation of our manuscript. As described below, we have revised our manuscript based on the reviewer’s worthy comments and recommendations.

Specific comments:

Comment 1:

Fig. 1/3/4/5/6: The presentation style of the chemical structures of the cholesterol-derivates is a bit unusual, because the structure is rotated. I understand that this might have been done in order to save space. However, it would look more familiar when the chemical structures are not turned and would be presented in the regular way. 

Response:

We agree with the reviewer’s suggestion and have corrected the structure of the molecules in Figures 1, 3, 4, 5 and 6.

Comment 2:

Fig. 1: This figure legend would benefit from one or more sentences of explanation.

Response:

Thank you for your suggestion. We have now elaborated the legends for Figure 1 (page 3).

Comment 3:

Conclusion: The authors have put together an interesting and informative review that could focus the attention of the readers more to the topic of RNAs in cancers. However, in the final sentences the authors may go a bit too far in terms of their wording. A review manuscript should not be called “in-depth research”, as it does not present original data. Moreover, the authors did not “validate” the correlation between specific RNAs and certain types of cancers, because they did not validate this experimentally but only discussed published data. Please rephrase this part.

Response:

As per the suggestion, we have now revised the concluding sentence (page 16, lines 552 and 553).

Additionally,

  1. The reference list has been modified and renumbered accordingly. Special attention is given to conform to the order of references and bibliographic style of the journal.
  2. The entire manuscript has been thoroughly checked and edited to ensure uniform style, organization, and quality.

On behalf of my co-authors, I once again express my sincere thanks to the erudite reviewer for the valuable suggestions and constructive input to improve the quality of our manuscript.

Reviewer 2 Report

The review manuscript entitled “MicroRNAs and Long non-coding RNAs as Novel Therapeutic Targets in Estrogen Receptor-Positive Breast and Ovarian Cancers” suggests the manipulation of ncRNA as a co-treatment scheme along other AI drugs on ER-positive patients of breast or ovarian cancers. The concept is not novel but a thorough review of the topic with updated bibliography could always produce an interesting work for a wide range of readers. I this direction is the inclusion of ovarian ER-positive cancers’ data.

The general parts (abstract-introduction) are quite well covered in the current manuscript but the rest of the text needs significant reconstruction. The methodology is highly non informative and results in very few candidate papers. As such, the resulted list of miRs and lncRNAs was limited and substantially reduced as compared to similar review articles of the past. It is surprising that even the current text contains more relevant citations than the ones included in Tables 1 and 2. The figures 3 and 4 would also need to be supplemented and corrected soas to clearly inform the reader about the mode of action of the relevant miRs in cancer, during treatment and their putative role as treatment targets. A suggested way to discriminate the putative candidates is to divide them according to their oncogenic and oncosuppressive potential and thus provide two separate figures accordingly. Also, the treatment’s effect on miR expression should be also highlighted, when applicable.  

Additional and more specific comments that would help the authors to reshape the current manuscript are following:

Abstract:

It is very detailed in some parts. The introduction seems to repeat the same level of details in the abstract or less (e.g. side effects on lines 21-22, list of miRs included in the study etc). Please consider either to reduce abstract’s details or add some more info in the introduction. Also, replace cancer by cancers in line 27.

Introduction:

The “ncRNA which include miRNAs and lncRNAs” has been repeatedly mentioned throughout the text. Omit this description from line 88 onwards.

Section 2:

It is very explanatory but at the same time not informative. It should include names of few representative databases utilized (lines 103-4), key word used for the search and the date when conducted the search (month, year). Replace “authors” by author if indeed there is only one (line 111). The text should include details on some parts of Fig. 2: What are the “other sources”? The duplicate removal is invalid since there were no duplicates (65+5=70). The inclusion/exclusion criteria and “the exclusion reasons”. How the n=14 was reduced to n=10 at the end is not justified.  

Section 3:

Lines 131-133 have been already introduced previously. Replace MMP by MMPs. Shibahara et al. with the very small clinical samples (n=3) should not be so highly considered. Lines 155-156 add appropriate reference. Line 159-160, add a couple of examples. Line 166, delete “a” from brackets. Line 173, replace lines by line. Lines 188 and 191, binds to instead of binds with. Line 190, provide the TF’s gene acronym. Line 198, omit “using 1cm tissue”. Bacci et al is repeatedly used in consecutive sentences in lines 217-223. Lines 222-223 omit female NCr foxhed nude”. GLUT-1 and TGFβR1 in lines 224 and 266 should be in italic. Line 243, omit “increased β-catenin”. MIR128-1, it is not clear the nature of the in vivo experiment with MCF7 cells and the absence of either patient samples or animal models in Table 1.

Table 1. Dosing or treatment could be added on the same column title next to “0cell lines used”.

Fig. 3 and 4. Is there a reason for being two separate figures? In all cases, the increase in miRs leads to increased estrogen irrespective of the up-/down-regulation of Aromatase? The relevance of the figure to the text is not very accurate in some cases (oncogenic and oncosuppressing miRs have to be overexpressed to increase ER???). The figures are not clear if depict the native effect of each miR or its overexpression on a certain mechanism. 

Section 4:

Lines 315-319. They repeat themselves, needs editing. Similar for lines 323-326 which circulate previously provided info. Remove “by” in line 379. Add “11” after “metalloproteinase” in line 381. Remove the first brackets and their text from line 392. Lines 397-399 are identical to lines 373-376.

Table 2. Row 1. According to the text, reference 74 refers to OPSC patient sample which is missing, KGN cell lines reported by reference 76 and reference 77 utilized 3 month old nulliparous rats as animal model.

Section 5:

Explain acronym TNBC in line 503.

Author Response

The authors of this manuscript express their sincere thanks to the reviewer for the critical assessment of this work. The authors have acted upon the recommendations of the reviewer which have resulted in a significant enhancement in the quality of this manuscript. All modifications incorporated in the manuscript are highlighted in red color font. A “point-by-point” response to each and every comment is outlined below.

General comments:

Comment 1:

The review manuscript entitled “MicroRNAs and Long non-coding RNAs as Novel Therapeutic Targets in Estrogen Receptor-Positive Breast and Ovarian Cancers” suggests the manipulation of ncRNA as a co-treatment scheme along other AI drugs on ER-positive patients of breast or ovarian cancers. The concept is not novel but a thorough review of the topic with updated bibliography could always produce an interesting work for a wide range of readers. I this direction is the inclusion of ovarian ER-positive cancers’ data.

Response:

We agree with the reviewer and we have added the ER-positive ovarian cancer in the revised manuscript (Section 4, page 11, lines 324-327).

Comment 2:

The general parts (abstract-introduction) are quite well covered in the current manuscript rest of the text needs significant reconstruction. The methodology is highly non informative and results in very few candidate papers. As such, the resulted list of miRs and lncRNAs was limited and substantially reduced as compared to similar review articles of the past. It is surprising that even the current text contains more relevant citations than the ones included in Tables 1 and 2.

Response:

As suggested, we have added references in Tables 1 and 2. Additionally, we had made substantial changes in the methodology section describing the searching database and various inclusion and exclusion criteria have been discussed in detail (page 3, line 144 to page 4, line 146; page 4, lines 149-155).

Comment 3:

The figures 3 and 4 would also need to be supplemented and corrected so as to clearly inform the reader about the mode of action of the relevant miRs in cancer, during treatment and their putative role as treatment targets. A suggested way to discriminate the putative candidates is to divide them according to their oncogenic and oncosuppressive potential and thus provide two separate figures accordingly. Also, the treatment’s effect on miR expression should be also highlighted, when applicable.  

Response:

Thank you for your thought-provoking comments and suggestions. We have now segregated results based on oncogenic and tumor-suppressive potential (page 6, lines 183 and 184; page 11, lines 353-354).

Specific comments:

Additional and more specific comments that would help the authors to reshape the current manuscript are following:

Response:

We have addressed the reviewer’s specific concerns as follows.

Comment 1:

Abstract:

It is very detailed in some parts. The introduction seems to repeat the same level of details in the abstract or less (e.g. side effects on lines 21-22, list of miRs included in the study etc). Please consider either to reduce abstract’s details or add some more info in the introduction. Also, replace cancer by cancers in line 27.

Response:

Thank you for your kind suggestion. We have now incorporated the changes and deleted redundant text in the abstract and introduction section as suggested (page 1, lines 20 and 21; page 2, lines 75-78).

Comment 2:

Introduction:

The “ncRNA which include miRNAs and lncRNAs” has been repeatedly mentioned throughout the text. Omit this description from line 88 onwards.

Response:

We agree with the suggestion and omitted the text “ncRNA (miRNAs & lncRNA)” in the revised manuscript (page 2, line 88).

Comment 3:

Section 2:

  1. It is very explanatory but at the same time not informative. It should include names of few representative databases utilized (lines 103-4), key word used for the search and the date when conducted the search (month, year).
  2. Replace “authors” by author if indeed there is only one (line 111).
  3. The text should include details on some parts of Fig. 2: What are the “other sources”?
  4. The duplicate removal is invalid since there were no duplicates (65+5=70).
  5. The inclusion/exclusion criteria and “the exclusion reasons”. How the n=14 was reduced to n=10 at the end is not justified.  

Response:

  1. a) We agree with the suggestion and incorporated the suggested information in the text (page 3, lines 102-104).
  2. b) We agree with the suggestion and replaced “authors” by author in (page 4, line 114).
  3. c) We agree with the suggestion and incorporated database information in text (page 3, lines 103-105).
  4. d) We agree with the suggestion and updated, corrected the mathematical error (page 4, lines 107-113).
  5. e) We agree with the suggestion and updated, “the exclusion reasons” in the text (page 4, lines 110-112).

Comment 4:

Section 3:

  1. Lines 131-133 have been already introduced previously. Replace MMP by MMPs. Shibahara et al. with the very small clinical samples (n=3) should not be so highly considered.
  2. Lines 155-156 add appropriate reference.
  3. Line 159-160, add a couple of examples.
  4. Line 166, delete “a” from brackets.
  5. Line 173, replace lines by line.
  6. Lines 188 and 191, binds to instead of binds with.
  7. Line 190, provide the TF’s gene acronym.
  8. Line 198, omit “using 1cm tissue”.
  9. Bacci et al is repeatedly used in consecutive sentences in lines 217-223.
  10. Lines 222-223 omit female NCr foxhed nude”.
  11. GLUT-1 and TGFβR1 in lines 224 and 266 should be in italic.
  12. Line 243, omit “increased β-catenin”.
  13. MIR128-1, it is not clear the nature of the in vivo experiment with MCF7 cells and the absence of either patient samples or animal models in Table 1.
  14. Table 1. Dosing or treatment could be added on the same column title next to “0cell lines used”.
  15. 3 and 4. Is there a reason for being two separate figures? In all cases, the increase in miRs leads to increased estrogen irrespective of the up-/down-regulation of Aromatase? The relevance of the figure to the text is not very accurate in some cases (oncogenic and oncosuppressing miRs have to be overexpressed to increase ER???). The figures are not clear if depict the native effect of each miR or its overexpression on a certain mechanism. 

Response:

  1. a) We agree with the suggestion and made replacement as per reviewers’ suggestions in text (page 5, lines 151; page 5, lines 153 and154).
  2. b) We agree with the suggestion and added the relevant references (page 5, line 159).
  3. c) We agree with the suggestion and added examples as per reviewers’ instructions (page 5, line 164).
  4. d) We agree with the suggestion and deleted “a” from brackets (page 6, line 189).
  5. e) We agree with the suggestion and replaced lines with sample (page 6, line 199)
  6. f) We agree with the suggestion and replaced “binds to” instead of “binds” (page 6, line 189)
  7. g) We agree with the suggestion and inserted “TF” full form (page 6, lines 188 and 189)
  8. h) We agree with the suggestion and omitted “using 1cm tissue” (page 6, line 200).
  9. i) We agree with the suggestion and omitted “increased β-catenin” (page 8, line 246)
  10. j) We agree with the suggestion and omitted “female NCr foxhed nude” (page 8, lines 264 and 265).
  11. k) We agree with the suggestion and italicized “GLUT-1” and “TGFβR1” (page 7, line 219; page 8, line 265).
  12. l) We agree with the suggestion and omitted “increased β-catenin” (page 8, lines 246).
  13. m) Revised the Table 1 as per the suggestions
  14. n) We agree with the suggestion unfortunately we could not find the dosing for all the molecules.
  15. o) We agree with the suggestion we have made substantial changes in the figures and separated the dysregulated ncRNA based on the oncogenic and tumor-suppressive potential.

Comment 4:

Section 4:

  1. Lines 315-319. They repeat themselves, needs editing.
  2. Similar for lines 323-326 which circulate previously provided info. Remove “by” in line 379. Add “11” after “metalloproteinase” in line 381.
  3. Remove the first brackets and their text from line 392.
  4. Lines 397-399 are identical to lines 373-376.
  5. Table 2. Row 1. According to the text, reference 74 refers to OPSC patient sample which is missing, KGN cell lines reported by reference 76 and reference 77 utilized 3 month old nulliparous rats as animal model.

Response:

  1. a) We agree with the suggestion and edited the text (page 11, lines 324-327).
  2. b) We agree with the suggestion and deleted “by” and added “11” in the final text (page 12, line 379).
  3. c) We agree with the suggestion and deleted the bracket and their text from line (page 12, line 392).
  4. d) We agree with the suggestion and made substantial changes in the section (page 12, lines 392-395).
  5. e) We agree with the suggestion and corrected the text in the manuscript (page 11, lines 341).

Comment 5:

Section 5:

Explain acronym TNBC in line 503.

Response:

Thank you very much. We have now added the full form of TNBC (page 15, lines 506 and 507). We have followed the rule of abbreviations throughout the manuscript.

Additionally,

  1. The reference list has been modified and renumbered accordingly. Special attention is given to conform to the order of references and bibliographic style of the journal.
  2. The entire manuscript has been thoroughly checked and edited to ensure uniform style, organization, and quality.

On behalf of my co-authors, I once again express my sincere thanks to the erudite reviewer for the valuable suggestions and constructive input to improve the quality of our manuscript.

Round 2

Reviewer 2 Report

I would like to thank the authors for carefully revising the manuscript and implementing almost all of my suggestions. I think that the revised manuscript offers an interesting and easy reading review.

I would like to add a couple of minor points and leave the implementation at author’s discretion.

  1. The initial Fig. 2 style could be followed in the revised Fig 2. I think that the former style and color provide a better illustration for to the reader.
  2. The subsections of sections 3 and 4 could be rearranged to follow the order of the revised Tables 1 and 2.
  3. The first “samples” in line 199 could be omitted.

Author Response

The authors of this manuscript once again express their sincere thanks to the reviewer for the critical assessment of this work. The authors have acted upon the recommendations of the reviewer which have resulted in a further improvement in the quality of this manuscript. All modifications incorporated in the manuscript are highlighted in a blue color font. A “point-by-point” response to each and every comment is outlined below.

General comments:

I would like to thank the authors for carefully revising the manuscript and implementing almost all of my suggestions. I think that the revised manuscript offers an interesting and easy reading review.

I would like to add a couple of minor points and leave the implementation at author’s discretion.

Response:

Thank you very much for your time and consideration. We are glad that the reviewer has pointed out the important corrections for the qualitative improvement of our manuscript.

Specific comments:

Comment 1:

The initial Fig. 2 style could be followed in the revised Fig 2. I think that the former style and color provide a better illustration for to the reader.

Response:

As suggested, we have revised the Figure 2 (page 3).

Comment 2:

The subsections of sections 3 and 4 could be rearranged to follow the order of the revised Tables 1 and 2.

Response:

Thank you for highlighting the above points. We have now rearranged the table as per the order of the sections 3 and 4 (Table 1 on page 6 and Table 2 on page 11).

Comment 3:

The first “samples” in line 199 could be omitted.

Response:

We have now omitted the first “Samples” and revised the sentence (page 6, lines 195 and 196).

On behalf of my co-authors, I once again express my sincere thanks to the erudite Assistant Editor and reviewer for the valuable suggestions and constructive input to improve the quality of our manuscript.